# Discovery and Mechanism of Novel 7-Aliphatic Amine Tryptanthrin Derivatives against Phytopathogenic Bacteria

**DOI:** 10.3390/ijms241310900

**Published:** 2023-06-30

**Authors:** Xuesha Long, Guanglong Zhang, Haitao Long, Qin Wang, Congyu Wang, Mei Zhu, Wenhang Wang, Chengpeng Li, Zhenchao Wang, Guiping Ouyang

**Affiliations:** 1School of Pharmaceutical Sciences, Guizhou University, Guiyang 550025, China; gs.xslong20@gzu.edu.cn (X.L.); gs.hylong21@gzu.edu.cn (H.L.); wq18385365070@163.com (Q.W.); gs.congyuwang20@gzu.edu.cn (C.W.); gs.mzhu20@gzu.edu.cn (M.Z.); whwang2302@163.com (W.W.); lichp11@163.com (C.L.); 2National Key Laboratory of Green Pesticide, Key Laboratory of Green Pesticide and Agricultural Bioengineering, Ministry of Education, Center for R&D of Fine Chemicals of Guizhou University, Guiyang 550025, China; glz2593195@126.com; 3Guizhou Engineering Laboratory for Synthetic Drugs, Guizhou University, Guiyang 550025, China

**Keywords:** 7-aliphatic amine tryptanthrin derivatives, *Xanthomonas oryzae* pv. *oryzae*, SEM, apoptosis

## Abstract

Rice bacterial leaf blight is a destructive bacterial disease caused by *Xanthomonas oryzae pv. oryzae* (*Xoo*) that seriously threatens crop yields and their associated economic benefits. In this study, a series of improved dissolubility 7-aliphatic amine tryptanthrin derivatives was designed and synthesized, and their potency in antibacterial applications was investigated. Notably, compound **6e** exhibited excellent activity against *Xoo*, with an EC_50_ value of 2.55 μg/mL, compared with the positive control bismerthiazol (EC_50_ = 35.0 μg/mL) and thiodiazole copper (EC_50_ = 79.4 μg/mL). In vivo assays demonstrated that **6e** exhibited a significant protective effect on rice leaves. After exposure, the morphology of the bacteria was partially atrophied by SEM. Furthermore, **6e** increased the accumulation of intracellular reactive oxygen species, causing cell apoptosis and the formation of bacterial biofilms. All the results indicated that **6e** could be a potential agrochemical bactericide for controlling phytopathogenic bacteria.

## 1. Introduction

*Oryza sativa* L. is a staple food crop, feeding more than half of the world’s population. However, it is often attacked by various pathogens, such as bacterial leaf blight in rice, during production, which seriously affects rice yield and quality and even endangers global food security and economic development [1,2,3]. *Xoo*, *Xanthomonas axonopodis* pv. *Citri* (*Xac*), and *Pseudomonas syringae pv. Actinidiae (Psa)* are the three major diseases threatening the health of agricultural crops. Among them, bacterial leaf blight is one of the three most serious rice diseases caused by *Xoo*, which harms the leaves and leaf sheaths of rice owing to its high epidemic potential and the lack of effective bactericides. In severe epidemics outbreaks, crop losses may be as high as 75%, and millions of hectares of rice are severely infected annually. Existing commercial antibacterials agents such as bismerthiazol (BT) and thiodiazole copper (TC) are incapable of effectively controlling the spread of bacterial diseases [4,5,6,7].

Moreover, the long-term, excessive use of traditional agrochemicals has led to imbalances of soil nutrients and a decline in soil fertility and organic matter, so the problem of soil and water pollution is becoming increasingly prominent. A large number of toxic substances’ residues also pose serious safety risks that threaten the safety of agricultural products and the environment. Therefore, driven by the importance of food security and increasing environmental protection, the creation of green pesticides with high efficiency, low toxicity, and low amounts of residue based on bio-derived pesticides is critical to solving traditional pesticide problems [8,9,10,11,12]. Natural compounds play an important role in drug discovery and are rich sources of lead compounds or pharmacophores used in new drug discovery. However, due to the limited resources of natural products, their low concentrations of active components, their poor specificity of action, and their inadequate pharmacokinetic properties, it is necessary to modify and optimize the structure of lead compounds via chemical synthesis to make them into more ideal drugs [13,14].

Tryptanthrin is a kind of indolequinazoline alkaloid that can be isolated from a variety of natural materials, including *Isatis tinctoria*, *Strobilanthes cusia*, *Polygonum tinctorium*, and *Wrightia tinctoria*. Tryptanthrin and its derivatives offer a wide range of biological activities, including antibacterial, anti-inflammatory, antileishmanial, antiviral, and antimalarial activity [15,16,17,18,19]. They also have excellent research and application prospects, as evidenced by their use in compounds such as Phaitanthrin A, Phaitanthrin B, and Phaitanthrin C (Figure 1A) [20,21,22,23]. Therefore, our group introduced novel 7-aliphatic amine of flexible fragments to the tryptanthrin master ring skeleton, expecting to obtain candidate compounds with stronger antibacterial activity (Figure 1B). Surprisingly, compound **6e** showed significant improvement against the three plant bacteria compared to the positive control drugs BT and TC, especially against *Xoo*. A valuable finding was that compound **6e** showed effective therapeutic and protective activity in vivo toward rice. The preliminary antibacterial mechanism of compound **6e** was also investigated using scanning electron microscopy (SEM), analyzing the accumulation of reactive oxygen species (ROS), examining rates of apoptosis, and through a biofilm assay. 

## 2. Results and Discussion

### 2.1. Chemistry

The titular compounds **6a**–**6z** were prepared using a series of optimized traditional methods (as depicted in Figure 1). In brief, the classic Sandmeyer reaction steps were adopted, wherein substituted anilines were employed as the starting materials. they reacted with chloral hydrate and hydroxylamine hydrochloride to yield the oxime group acetylaniline compound **2**, and the cyclization reaction was carried out with concentrated sulfuric acid, which was hydrolyzed to yield intermediates **3a**–**3i**. Meanwhile, various substituted isatoic anhydrides (**4a**–**4h**) were prepared via an oxidation reaction in dichloromethane solvent using *m*-chloroperbenzoic acid. Then, using acetonitrile as the solvent, using triethylamine as catalyst, the optimized Bergman method was used to perform reflux reaction and recrystallize methanol to obtain 7-chlorotryptanthrin derivatives containing various substituent groups, appearing yellow needle-shaped crystal compounds. Finally, through the traditional nucleophilic substitution reaction, aliphatic amines with a flexible structure were introduced into the tryptanthrin skeletons to obtain the target compounds **6a**–**6z**, which offer superior antibacterial activity against plant pathogens.

### 2.2. Biological Activity

#### 2.2.1. Antibacterial Activity of Target Compounds **6a**–**6z**

The activity of 7-aliphatic amine tryptanthrin derivatives was evaluated in vitro against three types of pathogenic bacteria using the turbidimetric method. Commercialized BT and TC served as reference substances. The resulting EC_50_ values are shown in Table 1; they suggest that most of the target compounds had prominent in vitro anti-*Xoo*, anti-*Xac*, and anti-*Psa* activity compared with the original skeleton of tryptanthrin and the control agents BT and TC. Notably, compound **6e** exhibited the strongest antibacterial activity, with EC_50_ values = 2.55 μg/mL and 4.01 μg/mL against *Xoo* and *Xac*, respectively. Compared to tryptanthrin, which presented EC_50_ values = 117 μg/mL and 126 μg/mL, amounting to 45.8-fold and 31.4-fold increases. Compared to BT, with EC_50_ values = 35.0 μg/mL and 53.7 μg/mL, which constituted 13.7-fold and 13.4-fold increases. Compared to TC, with EC_50_ values = 79.4 μg/mL and 68.9μg/mL, amounting to 31.1-fold and 17.2-fold increases.

Analysis of antibacterial activity. Combined with EC_50_ values in Table 1, the introduction of 7-aliphatic amine to tryptanthrin significantly increased antibacterial activity. For instance, when the 7-position substituent group was *N1*, *N1*-dimethylpropane-1,3-diamine, superior inhibitory activity toward *Xoo* was observed. The 2-substitution of tryptanthrin appears to be associated with higher antibacterial effects against *Xoo* and increased activity, presenting an (R_2_) of (**6e**) F > (**6i**) OCH_3_ > (**6g**) Br> (**6b**) H> (**6j**) CH_3_ > (**6k**) NO_2_. As a cycloaliphatic amine substituent, piperazine is often used as a linker to link active substructures with promising biological activities, allowing for the significant improvement of the antibacterial activity of the corresponding compounds. However, the inhibitory activity of methylpiperazine substituents was weaker than that of piperazine. When the substitution group was morpholine, the antibacterial activity was weaker and almost non-existent.

#### 2.2.2. In Vivo Bioassay Results of Compound **6e** against Rice Bacterial Leaf Blight

Given the superior in vitro bioactivity of compound **6e** against *Xoo* (EC_50_ = 2.55 μg/mL), in vivo bioassays concerning compound **6e** were further conducted, and the results are shown in Figure 2 and Table 2. Notably, compound **6e** presented better control efficacy with respect to anti-*Xoo* activity (45.02% of protective activity and 43.75% of curative activity) toward rice bacterial blight under controllable greenhouse conditions at a dosage of 200 μg/mL; this activity is superior to that of commercialized BT (42.18% of protective activity and 47.32% of curative activity). After bacterial infection, the degree of leaf wilt in the blank control group and the 6e treatment group changed, as shown in Figure 2. Most of the leaves in the untreated group presented chlorosis and withered leaves, while only a small amount of bacterial contamination occurred in the tips of the leaves in the treated group. Compound **6e** presented significant therapeutic and protective effects in the in vivo rice experiments. Therefore, it has desirable applicational prospects in the control of diseases caused by rice bacteria.

#### 2.2.3. Effect of Compound **6e** on the Morphology of Xoo Cells

In order to further verify the cells morphological changes induced by compound **6e** on the *Xoo* bacteria, SEM imaging was performed (Figure 3). At a compound **6e** concentration of 10 × EC_50_, it could be observed that part of the membrane of *Xoo* cells had assumed a folded, rod-shaped form and begun to shrink and collapse. When the concentration of compound **6e** was increased to 20 × EC_50_, the cell morphology began to show severe shrinkage and deformation. However, compared to the untreated *Xoo* cells, there were complete cell membranes and complete rod shapes. Especially in the areas marked by red arrows, the collapse and contraction of cells were clearly evident. These results suggested that higher concentrations of compound **6e** could affect the membrane integrity of *Xoo* cells and ultimately lead to bacterial apoptosis. The SEM results showed that compound **6e** had a concentration-dependent effect on cell morphology. Since there are many factors that affect the alteration of cell morphology, we studied their interactions in terms of the elevation of ROS, the rate of apoptosis, and the influence on the formation of bacterial biofilms.

#### 2.2.4. Compound **6e** Induced ROS Accumulation

On the one hand, reactive oxygen species (ROS) play an important role in maintaining important biological functions of cells; On the other hand, an excessive accumulation of ROS can inhibit normal growth and even promote the apoptosis of cells. Therefore, to explore whether the significant anti-proliferative activity of compound **6e** was related to ROS levels, *Xoo* cells were treated with **6e** for 12 h ((0, 1 × EC_50_, 2 × EC_50_, 5 × EC_50_, and 10 × EC_50_) and then loaded with 10 mM DCFH-DA for the examination of ROS using kit. Intracellular ROS can oxidize non-fluorescent DCFH to produce fluorescent DCF, and the fluorescence value of DCF can be detected to determine the levels of intracellular ROS. The results demonstrated that compound **6e** significantly increased intracellular ROS accumulation in a dose-dependent manner, especially at a concentration of 10 × EC_50_, which the amount of ROS reached the maximum value of 157; Even at a lower concentration of 1 × EC_50_, ROS also increased significantly (Figure 4). Therefore, it was concluded that the significant inhibitory effect of compound **6e** on *Xoo* pathogens may be related to the accumulation of intracellular ROS.

#### 2.2.5. Compound **6e** Induced Bacterial Cells’ Apoptosis

The accumulation of intracellular reactive oxygen species may eventually lead to cell apoptosis. Therefore, to further verify that **6e** induced apoptosis, *Xoo* cells were treated with either DMSO or various other concentrations of compound **6e**. Then, the cells were harvested and stained with Annexin V-FITC and propidium iodide (PI), and the percentage of apoptotic cells was analyzed using flow cytometry. As shown in Figure 5, the apoptotic rates of the drug-treated cells were positively correlated with the concentration changes. After treatment with different concentrations of 1 × EC_50_, 2 × EC_50_, 5 × EC_50_, and 10 × EC_50_, the total apoptotic rates were 18.76%, 22.53%, 32.87%, and 65.24%, respectively, whichcan be compared with that of 8.87% in the negative control group, thus showing that compound **6e** effectively induced apoptosis in *Xoo* cells.

#### 2.2.6. Compound **6e** Inhibited the Formation of Bacterial Biofilms

Biofilms are mostly composed of microcolonies encased in an EPS matrix, proteins, and extracellular DNA, which adhere to each other in very fine ways, forming membranes that act as barriers and space-occupying forms of protection, preventing foreign germs from colonizing the body and invading through the portal [24]. Therefore, through a crystal violet staining experiment, the OD value was measured at the 570 nm wavelength, and the biofilm was quantitatively measured to study whether compound **6e** could achieve an antibacterial effect by inhibiting the formation of *Xoo* bacterial biofilm; the corresponding results are shown in Figure 6. Obviously, the degree of biofilm formation was greater in the blank control sample, in which a bright blue biofilm was presented, and the maximum average OD value was 2.48. With the addition of compound **6e**, the degree of biofilm formation gradually decreased, and the bright blue color gradually became lighter. When the concentration increased to 10 × EC_50_, the color was at its lightest, and the minimum average OD value was 0.507. These results suggest that compound **6e** may act on *Xoo* bacteria in a dose-dependent manner by influencing the formation of bacterial biofilms.

## 3. Materials and Methods

### 3.1. Chemistry

The ^1^H and ^13^C NMR data were obtained using a Bruker Avance III 400 MHz NMR spectrometer (Bruker Optics, Billerica, MA, USA), for which CDCl_3_ was used as a solvent and the chemical shift (δ) was expressed in parts per million (ppm). High-resolution mass spectra (ESI-HRMS) were acquired using a Thermo Scientific Q Exactive Hybrid Quadrupole-Orbitrap mass spectrometer (Thermo Scientific, St. Louis, MO, USA). The melting points (mp.) for all the titular compounds were detected using a (X-4D) digital micro-melting point apparatus (Cewei Optoelectronic Technology Co., Ltd., Shanghai, China). The data on ROS and apoptosis were obtained using a BD FACSCalibur flow cytometer(Beckman Coulter, Inc. 250S. Kraemer Blvd. Brea, CA, USA). All the starting materials, reagents, and solvents were purchased from commercial sources without further purification. Detailed characterization data are presented in the Appendix A).

#### 3.1.1. General Synthesis Procedure for Intermediates **3a**–**3i** and **4a**–**4h**

The syntheses of intermediates **3a**–**3i** were conducted by referring to the literature [25,26,27]. Indoline-2,3-dione derivatives were synthesized using a lightly optimized procedure for provoking the Sandmeyer reaction, in which the intermediate compound **2** was obtained through a mixture of chloral hydrate, substituted anilines, hydroxylamine hydrochloride, hydrochloric acid, and sulfate-saturated aqueousa. Additionally, concentrated sulfuric acid was stirred in a heating system, with the temperature strictly maintained at 90 °C for 15 min, and then poured into ice water for the cyclic reaction for 30 min to obtain **3a**–**3i**. Compounds **4a**–**4h** were obtained via reflux reaction with 1.2 equivalents of *m*-chloroperoxybenzoic acid in dichloromethane solvent.

#### 3.1.2. General Synthesis Procedure for Intermediates **5a**–**5h**

Intermediates **5a**–**5h,** with various substituent groups of 7-chlorotryptanthrin, were prepared according to a previously reported method, which simple optimization was carried out [28,29].

#### 3.1.3. General Procedures for the Synthesis of Target Compounds **6a**–**6z**

Potassium carbonate (3.0 mmol, 0.414 g) was added in batches to a mixture of 7-chlorotryptanthrin and a set of its derivatives, namely, **5a**–**5h** (1.0 mmol, 0.282 g), in DMF (5.0 mL) and stirred at room temperature for 30 min. Then, the corresponding aliphatic amines (3.0 mmol) were added, and the resulting mixture was refluxed for 1–3 h. Finally, the precipitated solid was filtered and washed carefully with cold, distilled water and methanol to obtain the desired target compounds **6a**–**6z**.

### 3.2. In Vitro Antibacterial Bioassay

In vitro antibacterial activities of title compounds **6a**–**6z** against three pathogenic bacteria, namely, *Xoo*, *Xac*, and *Psa*, were determined using the classical turbidimetric method [30,31,32]. TC and BT were employed as representative commercial drugs serving as positive controls, and DMSO was used as the blank control. The inhibition rate (*I*) was calculated as follows: *I* (%) = (*C* − *T)*/*C* × 100. C is obtained by correcting the turbidity value of bacterial growth on the treated NB, and T is the drug-containing sample. The inhibitory activity of all target compounds against three strains of bacteria at different concentrations was further determined. The EC_50_ values were obtained by measuring the optical density value at 595 nm, and SPSS 19.0 software was used for statistics.

### 3.3. In Vivo Assay against Rice Bacterial Blight

The protective and curative activities of compound **6e** against rice bacterial leaf blight were determined using a previously reported method with some slight modifications [33]. “Fengyouxiangzhan” rice variety was grown until the tillering stage (28 °C and 90% RH) under greenhouse conditions to perform the antimicrobial experiment. After inoculation using a shear method, the therapeutic and protective effects of rice were tested using a spraying method. BT (90% white powder) and TC (20% suspending agent) were used as positive control agents. The rice plants cultured under controlled greenhouse conditions were subjected to in vivo experiments conducted on rice leaves at the tillering stage. The control efficiency was calculated using the following formula after 14 days of activity at a drug concentration of 200 μg/mL: *I* (%) = (*C* − *T*)/*C* × 100%. In the formula, *C* is the disease index for the negative control, and T is the disease index for the treatment group.

### 3.4. Morphological Investigations Using SEM

A bacterial morphology assay was carried out using previous research as a reference [34,35,36]. Briefly, 1.5 mL of *Xoo* cell suspension grown until the logarithmic stage was centrifuged, collected, washed with cooled PBS buffer solution 3 times, and then re-suspended with the same volume of PBS. Various concentrations of compound **6e** (0, 10 × EC_50_, and 20 × EC_50_) were added and incubated for 8 h at 28 °C/180 rpm. Then, the samples were washed with PBS 3 times once again. The *Xoo* cells were fixed overnight with 2.5% glutaraldehyde, dehydrated with gradient ethanol, replaced with tert-butanol solvent, and finally vacuum freeze-dried. Morphological changes of the *Xoo* bacteria were observed using scanning electron microscopy.

### 3.5. Detection of Reactive Oxygen Species

To assess the intracellular production of ROS, the *Xoo* pathogenic bacterial cells were treated with different concentrations of **6e** (0, 1 × EC_50_, 2 × EC_50_, 5 × EC_50_, and 10 × EC_50_) using a previously reported method with a slight modification [37]. After treatment for 12 h, cells were harvested and stained with 10 μM of DCFH-DA for 30 min at room temperature using a cell-based ROS assay kit. Next, cells were collected and measured using a BD FACSCalibur flow cytometer (Beckman Coulter, Inc. 250S. Kraemer Blvd., Brea, CA, USA).

### 3.6. Induction of Apoptosis in Pathogenic Bacterial Cells

To investigate whether the cell growth inhibition was associated with apoptosis, the plant pathogenic bacterial cells were treated with either DMSO or various concentrations of the target compound for 15 h using previous reported methods with slight modifications [38,39]. After treatment with different concentrations of **6e** (0, 1 × EC_50_, 2 × EC_50_, and 5 × EC_50_), the cells were washed twice with cold PBS, centrifuged, and collected. Then, the cells were stained with Annexin V-FITC and propidium iodide (PI) and analyzed using a flow cytometer to detect the apoptosis ratio. Quadrant panels were separated into four gates labeled Q1 (necrotic cells), Q2 (late apoptotic cells), Q3 (early apoptotic cells), and Q4 (viable cells). The control group consisted of untreated cells.

### 3.7. Bacterial Biofilm Assay

All of the strains of bacteria tested were provided by the Laboratory of Plant Disease Control at Guizhou University. NA media containing *Xoo* bacteria (OD_595_ = 0.2) and different concentrations of compound **6e** (0, 1 × EC_50_, 2 × EC_50_, 5 × EC_50_, and 10 × EC_50_) were prepared in 5 mL glass test tubes and then added to 96-well polystyrene plates, as previously described [40,41]. After being cultured in a constant temperature incubator (28 °C) for 3 days, the culture medium was removed and cleaned twice with sterile, secondary distilled water. Next, after being dried at room temperature, crystal purple solution (0.1%) was added for a 10 min staining period. The staining agent was discarded and cleaned twice with sterile secondary distilled water again. Finally, the absorbance OD value was measured at 570 nm after the sample was dried again and fully dissolved with 95% ethanol.

## 4. Conclusions

In this study, a series of 7-aliphatic amine tryptanthrin derivatives, **6a**–**6z**, was designed and synthesized by combining flexible small molecular fragments of aliphatic amines, which enhanced the solubility of the target compounds and broke the structures of planar aromatic molecules. The antibacterial bioassay results showed that most of the target compounds exhibited obvious antibacterial activity against three species of plant bacteria in vitro. Furthermore, compound **6e** displayed the greatest antibacterial activity against *Xoo* in vitro and was effective in reducing rice bacterial leaf blight in vivo. The preliminary mechanism of compound **6e** was investigated, and the results demonstrated that compound **6e** could precipitate obvious changes in bacterial morphology, induce ROS accumulation, stimulate cell apoptosis, and inhibit the formation of biofilms. The current work provides valuable insights into tryptanthrin and its derivatives for use in the field of natural agricultural bactericides.

## Data Availability

Not applicable.

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
