# Peer review of "Discovery and Mechanism of Novel 7-Aliphatic Amine Tryptanthrin Derivatives against Phytopathogenic Bacteria"

_ijms, 2023, doi:10.3390/ijms241310900_

Round 1
Reviewer 1 Report
Я дійсно згоден з твердженням авторів про те, що рис є основною продовольчою культурою, яка годує більше половини населення планети. І досить цікава наукова робота авторів!
Стаття структурована і містить повний обсяг інформації з обраної теми.
Єдине питання до авторів, рядок 189, чи покаже експеримент достовірний результат, якщо промити тверде тіло, яке відокремилося, звичайною холодною водою? Чи можна митися дистильованою водою
I really agree with the authors' statement that rice is the main food crop that feeds more than half of the world's population. And quite interesting scientific work of the authors! The article is structured and contains a full volume of information on the selected topic.
The only question for the authors, line 189, will the experiment show a reliable result if you wash the solid that has separated with ordinary cold water? Is it possible to wash with distilled water?

Author Response
Point 1: “The only question for the authors, line 189, will the experiment show a reliable result if you wash the solid that has separated with ordinary cold water? Is it possible to wash with distilled water?”
Response 1: Thanks for your suggestion. There may have been a misunderstanding due to our incomplete statement. However, in the actual synthesis experiment operation, we obtained the target compound after washing in pre-cooled distilled water. Therefore, in the manuscript we have fully described as "distilled water". Again, thank you for your valuable advice.

Reviewer 2 Report
Please consider adding a plate with photos with ROS production in the tissues of the plants studied.
For figure 2, please add magnifications of the leaves to illustrate the damage caused by the bacteria.
Add documentation to the article on how the substances affect the bacteria. The authors showed SEM documentation that would indicate that changes in the bacterial cell wall occur.
Add some Transmission Electron Microscope images to show what changes occur in the cells (cell wall, membranes, etc.).
Author Response
Point 1: “Please consider adding a plate with photos with ROS production in the tissues of the plants studied.”
Response 1: Thanks for your suggestion. We reviewed a lot of literature and tried to study the effect of compound 6e on ROS in rice leaves. Due to the current rice growth state and experiment period, the experiment could not get the ideal experimental results in a short time. Thank you very much for your valuable suggestions. In the later in-depth mechanism study, we will carefully analyze this experimental phenomenon and look forward to finding better experimental results.
Point 2: “For figure 2, please add magnifications of the leaves to illustrate the damage caused by the bacteria.
Response 2: Thanks for your suggestion. We have added the magnifications of the leaves of blank control and 6e on protective effect to Figure 2 in vivo, which are described in the manuscript as follows: The blank control group after rice leaf infection and the plant protection group of 6e were enlarged as shown in Figure 2. Most of the leaves in the untreated group showed chlorosis and withered leaves, while only a small amount of bacterial contamination occurred in the tips of the leaves in the treated group. Compound 6e has significant therapeutic and protective effects in rice experiments in vivo.
Point 3: “Add documentation to the article on how the substances affect the bacteria. The authors showed SEM documentation that would indicate that changes in the bacterial cell wall occur.”
Response 3: Thanks for your suggestion. The SEM documentation showed that the co-incubation of compound 6e with Xoo resulted in some changes in cell morphology. This may be related to ROS, cell apoptosis and affect the formation of biofilm, which is also confirmed in the later experimental results. Therefore, in manuscript we have added the description: The SEM results showed that compound 6e had a concentration-dependent effect on cell morphology. Since there are many factors that affect the change of cell morphology, we studied their interactions in terms of the elevation of ROS, the rate of apoptosis, and the influence on the formation of bacterial biofilm.
Point 4: Add some Transmission Electron Microscope images to show what changes occur in the cells (cell wall, membranes, etc.).”
Response 4: Thanks for your suggestion. We tried to further observe the morphological changes of the suborganelles of Xoo cells through transmission electron microscopy, but due to the small cell volume, we could only see the external morphology of the cells, which was similar to the result of scanning electron microscopy, but not more intuitive than the stereoscopic image of scanning electron microscopy. However, your valuable suggestions have provided us with new ideas for the in-depth study of the mechanism of action. In the later experiments, we will study in detail the degree of subtle changes in cell morphology observed by transmission electron microscopy. Thank you again for your careful consideration and suggestions on this manuscript.

Reviewer 3 Report
Please see the attachment

Minor editing of English language required
Author Response
Point 1: The section "2.1. Chemistry" suffers from the absence of any structure-activity discussion within the extended series of the compounds under study in the manuscript. A vast majority of the compounds appeared to possess much higher antibacterial activity as compared to tryptanthrin and the reference antibiotics bismerthiazol, thiodiazole copper. Such compounds deserve the SA discussion.
Response 1: Thanks for your suggestion. In manuscript, we put the discussion part of SA in “Part 2.2.1” “Analysis of antibacterial activity”. The results of the discussion are as follows: Combined with EC50 values in Table 1, the introduction of 7-aliphatic amines to Tryp significantly increased anti-bacterial activity. For instance, when the 7-position substituent group was N1, N1-dimethylpropane-1,3-diamine, showed superior inhibitory activity of Xoo. The 2-substitution of Tryp appears to be associated with higher antibac-terial effects of Xoo and increased activity with (R2) of (6e) F > (6i) OCH3 > (6g) Br> (6b) H> (6j) CH3 >(6k) NO2. Pi-perazine, as a cyclic adipose amine substituent, could significantly improve the activity of the compound against three strains of bacteria. The inhibitory activity of methylpiperazine substituents was weakened compared with piperazine. However, When the substitution group was morpholine, the antibacterial activity was weaker and al-most non-active.
Point 2: Figure 3: "SEM images for Xoo after incubated in 10×EC50 and 20×EC50 concentrations of compound 6e" seems not representative as for the effect of the compound 6e on the bacteria morphology. Either another image or simply the arrows running between the elements containing in the present image would be better.
Response 2: Thanks for your suggestion. In Figure 3, we have clearly marked the changes of cell morphology in SEM images after treatment of compound 6e with red arrows. Thank you for your advice on the SEM image processing.
Point 3: The section "3.2 In vitro antibacterial bioassay" should contain a somewhat more detailed description of the bioassay, not only Refs.
Response 3: Thanks for your suggestion. In manuscript, we added a detailed description of the in vivo experiments of rice, including the varieties of rice, the growing environment, and the methods of inoculation and drug application. “Fengyouxiangzhan” rice variety was grown to tillering stage (28 ºC and 90%RH) under greenhouse conditions for antimicrobial experiment. After inoculation by shear method, the therapeutic and protective effects of rice were tested by spraying method.
Point 4: Line 91: "Piperazine … could significantly improve the activity of the three strains of bacteria" should be rephrased to mean the activity of piperazine, but not of strains
Response 4: Thanks for your suggestion. There was a small error in the language description, which we have corrected and improved. “As a cycloaliphatic amine substituent, piperazine was often used as a linker to link active substructures with promising biological activities, which could significantly improve the antibacterial activity of target compounds.”
Point 5: Line 172-173: the section "3.1. Chemistry" contains non-chemical information on the strains of bacteria tested.
Response 5: Thanks for your suggestion. We have deleted the part of non-chemical information to make the manuscript information more reasonable. Thank you again for your careful consideration and suggestions on this manuscript.

Round 2
Reviewer 2 Report
I am satisfied with the authors' responses and changes to the article. I suggest to use TEM in future studies.